# Graphene Nanoribbon Bending (Nanotubes): Interaction Force between QDs and Graphene

Sahar Armaghani [1], Ali Rostami [1,*] and Peyman Mirtaheri [2,*]

1  Photonics and Nanocrystal Research Lab. (PNRL), Faculty of Electrical and Computer Engineering, University of Tabriz, Tabriz 51666-14761, Iran
2  Department of Mechanical, Electronics and Chemical Engineering, OsloMet—Oslo Metropolitan University, 0167 Oslo, Norway
*  Correspondence: rostami@tabrizu.ac.ir (A.R.); peyman.mirtaheri@oslomet.no (P.M.)

**Abstract:** Carbon materials in different shapes—such as fullerene molecules (0D), nanotubes and graphene nanoribbons (1D), graphene sheets (2D), and nanodiamonds (3D)—each have distinct electrical and optical properties. All graphene-based nanostructures are expected to exhibit extraordinary electronic, thermal, and mechanical properties. Moreover, they are therefore promising candidates for a wide range of nanoscience and nanotechnology applications. In this work, we theoretically studied and analyzed how an array of quantum dots affects a charged graphene plate. To that end, the array of quantum dots was embedded on the graphene plate. Then, considering the interaction between QDs and graphene nanoribbons, we transformed the charged plate of a graphene capacitor into a nanotube using the bipolar-induced interaction and the application of an external electromagnetic field. In this work, the dimensions of the graphene plate were 40 nm × 3100 nm. The bending process of a charged graphene plate is controlled by the induced force due to the applied electromagnetic field and the electric field induced by the quantum dots. Finally, using the predetermined frequency and amplitude of the electromagnetic field, the graphene nanoribbon was converted into a graphene nanotube. Since the electrical and optical properties of nanotubes are different from those of graphene plates, this achievement has many practical potential applications in the electro-optical industry.

**Keywords:** nanoribbon graphene; carbon nanotubes; array of quantum dots; rolling; dipolar interaction

## 1. Introduction

Since the shorter bond length of nanomaterials makes the materials stiffer and stronger, they show distinct properties [1,2]. One of the strongest bonds is carbon–carbon in the hexagonal lattice observed in solids [3,4]. Various allotropes of carbon nanomaterials exhibit each possible dimensionality, such as fullerene molecules (0D), nanotubes and graphene ribbons (1D), graphite plates (2D), nanodiamonds (3D), etc. Although the electronic structure of these allotropes (except for sp3 diamond) is similar to that of graphene (i.e., a complete unbounded single layer of sp2-bonded carbon atoms densely packed into a benzene ring structure), confinement effects play a crucial role. The lateral confinement of charge carriers, depending on the width of the ribbon, the nanotube diameter, and the stacking of the carbon layers with respect to the different crystallographic orientations involved, could create an energy gap near the Dirac point. Doping and topological defects (including edge disorders) have also been proposed as tools to tailor the quantum conductance in these materials after reviewing the transport properties of defect-free systems [5–7]. However, because planar graphene is unstable with respect to the formation of curved structures—such as fullerenes and nanotubes—it had been considered not to exist in a free state. Nevertheless, recently, researchers went on to construct graphene by mechanical exfoliation (i.e., repeated peeling or micromechanical cleavage) of bulk graphite (i.e., highly oriented pyrolytic graphite (HOPG)) [8,9] or by epitaxial growth through thermal decomposition of SiC [10]. However, the request for increasing the stability of the building element

without an increase in the size of the element leads to the introduction of reinforcements. This means that the structure of a single graphene sheet is more stable than that of other reinforcements. Although creating a graphene structure with other carbon allotropes means a less stable structure, with the advancement of technology and manufacturing theory, it is now possible to create a graphene sheet via modern methods, such as chemical vapor deposition (CVD). Thus, it is possible to make other allotropes without increasing their size, reducing them to micron and even nano size, and using graphene sheets. The use of carbon nanotubes (CNTs) for increasing the stability of materials was successfully carried out in the last decade of the 20th century [11]. CNTs have played an important role in the field of micro- and nano-development since their discovery, because of their outstanding conductivity, stiffness, and high aspect ratio. Sumio Iijima, in his work [12], discovered CNTs to be tubular carbon allotropes of graphite. The basic building block in nanotubes relies on the theoretical significance of the graphene sheet [5,13]. CNTs, depending on the number of graphene walls in their structure, can be categorized in two ways: single-walled CNTs (SWCNTs) and multiwalled CNTs (MWCNTs). The SWCNT is patterned to be a simple graphene sheet rolled to form a tubular structure. Nevertheless, an MWCNT is a group of several concentric rolled graphene sheets with a wall separation of ~0.34 nm [14]. Because of being held together by a delocalized electron cloud along the walls due to the sp2 hybridization of carbon atoms, MWCNTs are less flexible and have more structural defects compared to SWCNTs [15]. Moreover, Chang et al. found that upon rolling a graphene sheet for forming a chiral SWCNT, in-plane isotropy is maintained in the SWCNT [16]. The selection of employing MWCNTs or SWCNTs mostly depends upon the sphere of application. Most biomedical applications prefer SWCNTs, especially for drug delivery applications, because of their efficient drug-loading capacity and ultrahigh surface area. Engineering structures also use SWCNTs, because of their high stability compared to MWC-NTs [11]. Since the discovery of carbon nanotubes in 1991 by Ijima et al., some researchers around the world have extensively studied these structures. The large length of carbon nanotubes (up to several microns) and their small diameter (several nanometers) make the length-to-width ratio of these structures very high. The electronic, molecular, and structural properties of carbon nanotubes are largely determined by their nearly one-dimensional structure. Carbon nanotubes are produced in three main ways: arc discharge, laser ablation, and chemical vapor deposition. Nevertheless, scientists are exploring more economical ways to produce these structures. The main problem in the application of nanotubes is their high synthesis as well as their neutralization. Many applications of nanotubes are due to the intrinsic dimensions of nanoparticles, their high surface-to-volume ratio, and the unique combination of their electrical, optical, thermal, and structural properties. A.R. Kohler et al. reported some of the expected applications of nanotubes [17]. Carbon nanotubes are used as fillers in polymer composites to increase the electrical properties, tensile strength, durability, and conductivity of materials. Carbon nanotubes have been used as microelectrodes in polyvinylidene fluoride (PDVF) composites [18]. They can also potentially be used as electromagnetic interference (EMI) shields, artificial muscle, superconductors [19,20], hydrogen storage [21,22], fuel cells [23], fire-retardant sensors [24,25], and field emitters [26]. On the other hand, graphene is a compound that exhibits unusual physical properties that could be used in the future for electronics and optoelectronics. Although several methods have been developed for the synthesis of graphene, the control of process parameters is required to adapt the measurable energy gap with reproducible properties [27]. Kim et al. reported carrier mobility of slightly more than 200,000 cm$^2$/Vs for a single layer of mechanically exfoliated graphene. In addition, they specifically minimized the substrate-induced scattering by etching under the channel to produce graphene suspended between gold contacts in their experiments [28]. At room temperature, and with such high carrier mobility, charge transport is essentially ballistic on the micrometer scale. Because this enables the fabrication of all-ballistic devices even at current integrated circuit (IC) channel lengths (as low as 45 nm), it has great significance for the semiconductor industry. Graphene possesses considerable optical properties, and it can be optically imag-

ined, despite being only a single atom thick. The linear dispersion of the Dirac electrons makes broadband applications possible. Storable absorption is observed as a consequence of Pauli blocking, and the absence of equilibrium carriers results in hot luminescence. Chemical and physical treatments can also lead to luminescence. All of these properties make graphene an ideal photonic and optoelectronic material [27]. Chemical doping and defects in graphene-based materials are currently being actively explored as a potential source of innovation to tailor the electronic properties of these nanostructures. Recently, experiments on graphene have been extended to the fabrication and study of QD–graphene nanostructures [29,30]. The author of [30] observed that when an array of quantum dots placed on the modified Graphene, the graphene plate bends. In this work, we analyzed the effects of the polarization of the quantum dots' electric field on the graphene nanoribbon, which caused this bend. Moreover, we converted a graphene nanoribbon into a graphene nanotube by applying electromagnetic waves. This technique helps to use one material in two distinct structural states in electro-optical devices.

During the literature survey, we observed that the free vibration analysis of FG-CNTRC beams using higher-order zigzag theory (HOZT) remains untouched. Moreover, the modal stress analysis of FG-CNTRC beams under hygrothermal conditions was not explored fully. The free vibration behavior of FG-CNTRC beams under moisture conditions also awaits exploration. Within the study, free vibration analysis of FG-CNTRC beams was applied under hygrothermal conditions using recently proposed finite element (FE)-based HOZT. Hamilton's principle was applied for determining the frequencies of the FG-CNTRC beams under hygrothermal conditions. Temperature- and moisture-dependent material properties were used. The influence of the volume fraction of CNT, moisture concentration, and temperature on the free vibration and modal stress behavior of FG-CNTRC beams was tested well. Therefore, upon boosting the behavior of those microstructures, the material properties of nano-elements might change from one location to another, almost like functionally graded materials (FGMs). Sandwich construction is employed widely for constructing various structures within the aeronautics, aerospace, naval, automobile, and civil fields, among others [31–33]. For example, because transverse, commonplace stresses are taken into consideration in [31], functionally graded carbon-nanotube-reinforcement (FG-CNTR) was applied to bending and free vibration analysis. Moreover, at joints, the computational model accumulates transverse shear stress and transverse normal stress continuously. In fact, the zero transverse shear stress condition at the underside and top surfaces of the beam is also satisfied. As a result of the minimum energy, bending analysis was completed, while Hamilton's elements were adopted for free vibration analysis. The influence of the core's thickness on stresses and displacements was additionally critically analyzed. It was observed that the thickness of the core and the CNT gradation law significantly affect the mechanical behavior of the sandwich FG-CNTRC beam. The authors of [32] aimed to conduct free vibration analysis of functionally graded single-walled carbon-nanotube-reinforced composite (FG-CNTRC) beams under such conditions. Similar to reference [31], C-0 finite-element-based higher-order zigzag theory was administered as well, and was applied to five different graded CNTRC beams. Hamilton's principle was applied to outline the governing differential equation. Because of taking constant temperature or moisture distribution across the thickness of the beam, moisture-dependent material properties and temperature were used. Moreover, the modal stresses of the six primary mode shapes were studied. Modal stresses were found to worsen with temperature or moisture concentration as compared to the stresses observed for the essential mode of vibration. The character of stress distribution across the beam is determined by the gradation law together with the moisture or temperature values to which the beam is subjected. The FG-O beam had the lowest sensitivity under thermal conditions, whereas the FG-X beam had the highest. On the other hand, these nanomaterials are made up of individual units between 1 and 100 nanometers in size. Furthermore, they offer a lot of potential in many fields, such as pharmacy and biomedicine, owing to their exceptional physicochemical properties arising from their high surface area and nanoscale size, because

of which they have lately attracted a lot of attention. Smart engineering of nanostructures through appropriate surface or bulk functionalization bestows them with multifunctional capabilities and modern applications in the biomedical field, such as biosensing, drug delivery, etc. [34–36].

This paper is arranged into five sections: In the Section 2, we briefly introduce the geometry of SWCNTs and the crystallography of converting graphene nanoribbons to nanotubes. In the Section 3, our theory explains the cause of polarization of quantum dot arrays, which act as the electrostatic gate for graphene nanoribbons. This polarization exerts a force on the graphene nanoribbon sheet and causes it to bend. Moreover, the graphene nanoribbon plate bends to an initial point. Next, we examined how the quantum dots were positioned correctly, the maximum and minimum distances of the quantum dots from one another, and their number, so that by choosing the right hinge, the bending to be tubed could be helped. Thus, we used electromagnetic waves to bend the nanoribbons from the initial point to their endpoint, producing the structure of the graphene nanotubes, and in this section, methods to apply the waves, amplitude size, and radiation angle are calculated and expressed. In the Section 4, we present the results obtained. Finally, we discuss the importance of converting graphene sheets to graphene nanotubes in nanoelectronic and optical structures in Section 5.

## 2. Geometry and Crystallographic Structure of SWCNTs

Carbon nanotubes can be classified into MWCNTs and SWCNTs. An MWCNT consists of two or more concentric cylindrical cells of graphene sheets collocated coaxially around a central hollow, with interlayer separation, as in graphite (0.34 nm). However, an SWCNT is made of rolled single-layer graphene. The ends of the rolled single graphene sheet are sealed using two half-fullerene caps. The diameter of SWCNTs may vary from 0.7 to 2.5 nm. Originally, unlimited numbers of nanotube geometries would exist, because a graphene sheet would be rolled up at different angles. Different rolling angles result in different chirality or helicity of SWCNTs. Carbon atoms are arranged in the form of a honeycomb lattice in a graphene sheet. The lattice cell is defined with two primitive vectors α and β (see Figure 1).

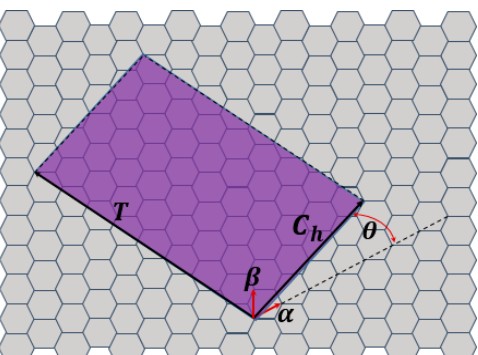

**Figure 1.** Schematic of an unrolled graphene sheet, determinations of geometrical parameters used to qualify a carbon nanotube, and forming an SWCNT with them [11,16].

A common approach is using a chiral vector or chiral angle to identify an SWCNT. The chiral vector $C_h$ in the graphene sheet can be indicated as a combination of primitive vectors. $C_h$ is defined as follows:

$$C_h = n\alpha + m\beta \equiv (n, m) \tag{1}$$

where $n$ and $m$ are the integers. When written as $(n, m)$, they are called a chiral index. If the head of the vector $C_h$ touches its tail when the graphene sheet is rolled into a tube, we call $C_h$ the chiral vector or roll-up vector of the nanotube [16]. Zigzag and armchair tubes are chiral nanotubes because of their high geometric symmetry. The angle $\theta$ is used for

determining the electrical properties of the SWCNT, and is referred to as the chiral angle (see Figure 1). However, SWCNTs with a chiral angle of $0 < \theta < \pi/6$ are chiral nanotubes. The chiral angle is given as follows:

$$\theta = \cos^{-1}\left(\frac{C_{h.\alpha}}{|\alpha||C_h|}\right) = \cos^{-1}\left(\frac{2n+m}{2\sqrt{n^2+m^2+nm}}\right) \tag{2}$$

The SWCNTs are of the following three types [11]:

- Armchair (A) SWCNT $n = m$, $C_h = (n, n)$, $\theta = \pi/6$.
- Zigzag (Z) SWCNT $m = 0$, $C_h = (n, 0)$, $\theta = 0$.
- Chiral SWCNT $n \neq m \neq 0$, $0 < \theta < \pi/6$.

Another important geometrical parameter of SWCNTs is the translation vector T, directed along the SWCNT axis and perpendicular to the chiral vector $C_h$ (see Figure 1). The diameter of SWCNTs can be determined from the following relation:

$$R = \frac{|C_h|}{\pi} = \frac{a_{c-c}}{\pi}\sqrt{3(m^2+n^2+mn)} \tag{3}$$

where $a_{c-c}$ displays the bond length between adjacent carbon atoms of a cell ($a_{c-c} \approx 1.42\ A$) [11].

We can conclude in this section that in order to create an armchair nanotube from a sheet of graphene nanoribbon, the parameters must be $n = m$, $C_h = (n, n)$, and $\theta = \pi/6$, so that the vector T is on an axis that is in line with the length and will pass through the center of the sheet of graphene nanoribbon. We deem it important that the presence of hanging bonding on the edge of the graphene nanoribbon sheet creates the final bond and cross-sectional formation. The cross-sectional radius is for the SWCNT. According to Formula (3), $n$ must be a non-integer value to create a hanging bond at the edge of the sheet. Thus, in our work, a width of 40 nanometers was determined to be sufficient for the graphene nanoribbon.

## 3. Mathematical Formalism

### 3.1. Interaction between Quantum Dots and the Charged Graphene Sheet

An electrostatic gate consists of two metal sheets in parallel and at an appointed distance, which are stuffed with a dielectric material, and these sheets are connected to a voltage source. We used a graphene nanoribbon sheet with an array of quantum dots as a metal layer (see Figure 2). The dimensions of the graphene nanoribbon sheet were 775 nm × 40 nm, on which an array of quantum dots with a dot radius of 10 nm was placed, with a permittivity of 12. A suitable voltage was attached to the graphene nanoribbon sheet to keep the chemical potential at 0.3 volts and to charge the graphene nanoribbon sheet [37]. According to nanomaterial technology, graphene can be grown on dielectrics such as $SiO_2$.

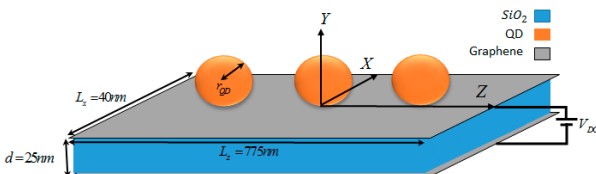

**Figure 2.** Schematic of the graphene–QD electrostatic gate structure.

An inactive graphene sheet has a chemical potential located at the Dirac point; the charge-carrier density at this point is zero, and does not act as a metal with unique properties in the electro-optical field. In the electrostatic gate structure, graphene sheets are charged after the voltage is applied. Furthermore, their chemical potential is removed from the Dirac point [31]. In graphene, the Fermi level ($E_f$), also known as chemical potential ($\mu_c$), has a finite value between 1 and −1 eV. For an electrostatic gate, whenever a DC voltage is connected to it, the charge stored inside the electrostatic capacitor is equivalent to the doped charge state in the graphene gate, which is proportional to the density of the charge

carriers at the Fermi level. The density of carriers at the Fermi level can be reduced to the following form through the Fermi–Dirac distribution and the density of two-dimensional material states [38,39]:

$$n_s = \frac{2}{\pi\hbar^2 v_f^2} \int_0^\infty E\left[f_d\left(E - E_f\right) - f_d\left(E + E_f\right)\right] dE = \frac{1}{\pi}\left(\frac{\mu_c}{\hbar v_f}\right)^2 \tag{4}$$

where $f_d$ is the Fermi–Dirac distribution, which is defined by the Fermi level ($E_f$) and the Boltzmann constant (KB = 8.617 × $10^{-5}$ ev·$K^{-1}$). An electron moves at a velocity equal to 106 m·$s^{-1}$ in a graphene structure, known as Fermi velocity ($V_f$). $\hbar$ is a symbol of Planck's decline ($\hbar$ = 6.5875 × $10^{-16}$ eV). Thus, the graphene sheet can be charged as much as $Q = q \cdot n_s$, where $q$ is the amount of electric charge ($q$ = 1.6 × $10^{-19}$ C).

As shown in Figure 3, this charged sheet creates an electric field around itself, resulting in the polarization of the QDs.

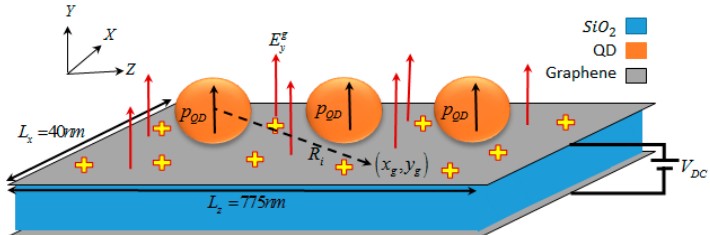

**Figure 3.** Illustration of the electric field emitted from the charged sheet of graphene nanoribbon and the polarization of QDs.

The coordinate position of each quantum particle is ($x_Q$, $z_Q$). If we consider that the charged sheet of graphene nanoribbon with dimensions of 775 nm × 40 nm is on the XZ plane, we can calculate this field using Gauss'law [40]. Therefore, the electric field emitted by a charged sheet of graphene nanoribbon at the center of the quantum dots on it, which polarizes the quantum dots, is equal to:

$$E_y^g = \frac{q.n_s}{4\pi\varepsilon_0 y^2 \left(\left(\frac{L_x}{y}\right)^2 + \left(\frac{L_z}{y}\right)^2 + 1\right)^{\frac{1}{2}}}, \qquad y = r_{QD} \tag{5}$$

The electric field emitted from the charged sheet of graphene nanoribbons leads to the polarization of quantum dots with $\varepsilon_r$ permittivity. The polarization measure of quantum dots is given as follows:

$$\begin{aligned} \vec{P}_0^{QD} &= \varepsilon_0(\varepsilon_r - 1)E_y^g a_y \\ \widetilde{p}_0^{QD} &= \frac{|q|(\varepsilon_r - 1)}{\left(\left(\frac{L_x}{r_{QD}}\right)^2 + \left(\frac{L_z}{r_{QD}}\right)^2 + 1\right)^{\frac{1}{2}}}\left(\frac{\mu_c}{2\pi r_{QD}\hbar v_f}\right)^2 \end{aligned} \tag{6}$$

According to Columbus's law, this creates an electric field around a polarized particle that is proportional to the amount of polarization and the distance from the center of polarization to the desired point '$p$' ($E = [3 (R \cdot P) R / R^2 - P]/4\pi\varepsilon_0 R^3$). Thus, the total electric field induced by the polarized particles to the desired point '$P$' with coordinate position ($x_p$, $y_p$, $z_p$) is given as follows:

$$E^{arry \ Dipolar} = \frac{3(y_p - r_{QD})\vec{p}_0^{QD}}{4\pi\varepsilon_0}\sum_{i=1}^{n}\frac{1}{R_i^3}\left[\frac{(x_p - x_{Q_i})}{R_i^2} \quad \left(\left(\frac{y_p - r_{QD}}{R_i^2}\right) - \frac{1}{3(y_p - r_{QD})}\right) \quad \frac{(z_p - z_{Q_i})}{R_i^2}\right]\begin{bmatrix} \cos\varphi & -\sin\varphi & 0 \\ \sin\varphi & \cos\varphi & 0 \\ 0 & 0 & 1 \end{bmatrix}\begin{bmatrix} a_r \\ a_\varphi \\ a_z \end{bmatrix} \quad (7)$$

$$R_i = \sqrt{(x_p - x_{Q_i})^2 + (y_p - r_{QD})^2 + (z_p - z_{Q_i})^2}, \cos\varphi = \frac{x_p}{\sqrt{x_p^2 + (y_p - r_{QD})^2}}, \sin\varphi = \frac{y_p - r_{QD}}{\sqrt{x_p^2 + (y_p - r_{QD})^2}}$$

where $R_i$ is the distance from the polarization center of each quantum dot to the desired point '$P$'. The electric field induced by quantum polarized particles creates an interaction between the quantum array and the charged sheet of graphene nanoribbon. The induced electric field exerts a force on the charged graphene sheet. Therefore, this force is proportional to the induced electric field.

$$\vec{F} = q \cdot n_s \cdot \vec{E}^{arry \ Dipolar} \rightarrow \begin{cases} F_r \propto E_r \\ F_\varphi \propto E_\varphi \\ F_z \propto E_z \end{cases} \quad (8)$$

*3.2. Analysis and Evaluation of Torque and Bending Force to Determine the Number and Arrangement of Quantum Dots on the Charged Sheet of Graphene Nanoribbon*

It should be noted that we wanted to bend a graphene sheet. The sheet bent due to the applied force on and the selected pivot for its torque. Here, we selected two modes of pivot: the first case of the pivot was a point in the center of the sheet (Figure 4a), and the second case of the pivot was axial along the length that passes through the center of the sheet (Figure 4b). In Figure 4, the green dots represent the points that are pivots. Since our purpose was to create a sphere from the first state and a cylindrical tube from the second state, we determined the direction of the force applied accordingly.

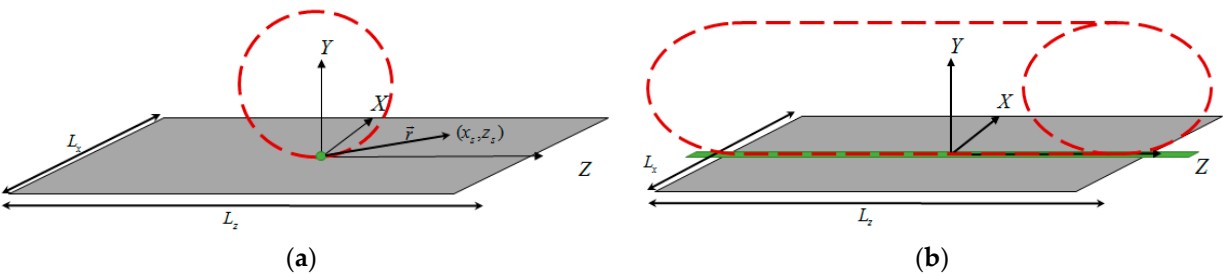

(a)　　　　　　　　　　　　　　　　　　　　　　　　　　　　(b)

**Figure 4.** Schematic of the pivot in two positions: (**a**) in the center of the sheet, and (**b**) on the axis passing through the center along the length. The pivot is represented in green.

We present the distance of each point from the sheet to the pivot by the r vector, and F is the vector of force that is applied to the sheet. Therefore, the torque applied to the sheet is given by ($\tau = r \times F$) [41]. In the first case, the distance vector is equal to $r = x_s \cdot a_x + z_s \cdot a_z$. For us to create the sphere from a flat sheet, the force and torque had to be applied to the flat sheet in all three directions of the coordinate system to obtain the desired shape. Thus:

$$\tau_z = x_s\left(F_r \sin\varphi + F_\varphi \cos\varphi\right)$$
$$\tau_r = -\left(z_s F_\varphi + x_s F_z \sin\varphi\right) \quad (9)$$
$$\tau_\varphi = z_s F_\varphi - x_s F_z \cos\varphi$$

where the angle $\varphi$ represents the angle of each point from the sheet to the center of the pivot, and each variable is with time. The same trend can be calculated for the second case. In the second case, the distance vector is equal to $r = x_s \cdot a_x$. In this case, there should be no torque to create a cylindrical tube along the length. As a result:

$$\tau_z = x_s\left(F_r \sin\varphi + F_\varphi \cos\varphi\right) = 0 \rightarrow \tan\varphi = -\frac{F_\varphi}{F_r}$$
$$\tau_r = -x_s F_z \sin\varphi \quad (10)$$
$$\tau_\varphi = -x_s F_z \cos\varphi$$

We wanted to use this technique to create carbon nanotubes from a charged sheet of graphene nanoribbon. The location of the quantum dot became the center of the pivot for torque. The purpose was to determine the number of quantum dots and how they were arranged. Based on this, it was clear that the quantum dots were aligned on the pivot axis (Figure 4b). Each quantum dot, depending on its position, can apply a torque proportional to Formula (9) on a charged sheet of graphene nanoribbon. This is important to create a graphene nanotube, as shown in Figure 4b. The torque on the pivot axis must be zero, so all of the torque values entered as the result of Equation (9) must be zero as well. The desired point 'p' on the axis has the coordinates $(0, 0, Z_p)$, so according to Formula (7), $R_i = \sqrt{r_{QD}^2 + (z_g - z_{Q_i})^2}, \cos \varphi = 0, \sin \varphi = -1$. In this case, according to Equation (8) and Formula (7), we have the following:

$$
\begin{aligned}
F_r &\propto E_r = \frac{\widetilde{p}_0^{QD}}{4\pi\varepsilon_0} \sum_{i=1}^{n} \frac{1}{R_i^3} \\
F_\varphi &\propto E_\varphi = 0 \\
F_z &\propto E_z = \frac{3r_{QD}\widetilde{p}_0^{QD}}{4\pi\varepsilon_0} \sum_{i=1}^{n} \frac{(z_g - z_{Q_i})}{R_i^5}
\end{aligned}
\tag{11}
$$

According to Formula (9), the distance from the pivot axis—the multiplied element with force in the z-direction—is zero, so there is no need to zero this force. However, to zero the torque values, the only element that must be zero is the force in the r-direction. As a result, Equation (12) is given for two consecutive quantum dots, as follows:

$$
\frac{\widetilde{p}_0^{QD}}{4\pi\varepsilon_0} \sum_{i=1}^{n} \frac{(z_g - z_{Q_i})}{R_i^3} = 0 \quad \rightarrow \quad \frac{(z_g - z_{Qi-1})}{R_{i-1}^3} = -\frac{(z_g - z_{Qi})}{R_i^3}
$$

$$
\frac{(z_g - z_{Qi-1})}{r_{QD}} \left(1 + \left(\frac{z_{Qi} - z_g}{r_{QD}}\right)^2\right)^{\frac{3}{2}} = \frac{(z_{Qi} - z_g)}{r_{QD}} \left(1 + \left(\frac{z_g - z_{Qi-1}}{r_{QD}}\right)^2\right)^{\frac{3}{2}}
\tag{12}
$$

If $Z_g$ is at the center point of the distance between the two quantum dots ($z_g = \frac{z_{Qi} + z_{Qi-1}}{2}$), Equation (12) is correct. Moreover, the force in the r-direction is zero.

Next, we relocate $Z_g$ from the center point of the distance between the two quantum dots to the edge of one of the quantum dots ($z_g = \frac{z_{Qi} + z_{Qi-1}}{2} \pm \Delta_{\max}$). The maximum distance that two quantum dots can have from one another, from edge to edge, is denoted by $2\Delta_{\max}$. Similarly, as shown in Figure 2, we define $\Delta$ as two quantum dots' distance from edge to edge—not from center to center—which is given by $\Delta = \left|\frac{z_{Qi} - z_{Qi-1}}{2}\right| - r_{QD}$. Thus, $\Delta_{\max}$ is $\left|\frac{z_{Qi} - z_{Qi-1}}{2}\right| - r_{QD}$ as well. To be able to solve Equation (12), we use variable change ($u = \frac{z_g - z_{Qi-1}}{r_{QD}}, g = \frac{z_{Qi} - z_g}{r_{QD}}$). Because $Z_g$ is on the edge of one of the quantum dots ($z_g = \frac{z_{Qi} + z_{Qi-1}}{2} + \Delta_{\max}$), '$u$' and '$g$' are $\frac{z_{Qi} - z_{Qi-1}}{r_{QD}} + 1$ and 1, respectively. Therefore, according to this trend, Equation (12) becomes:

$$
u^6 + 3u^4 - 5u^2 + 1 = \left(u^2 - 1\right)\left(u - 2 - \sqrt{5}\right)^2 \left(u - 2 + \sqrt{5}\right)^2 = 0
$$

$$
u = \frac{z_{Qi} - z_{Qi-1}}{r_{QD}} + 1 =
\begin{cases}
1 & no \\
-1 & no \\
2 - \sqrt{5} & no \\
2 + \sqrt{5} & yes \rightarrow z_{Qi} - z_{Qi-1} = (1 + \sqrt{5})r_{QD}
\end{cases}
\tag{13}
$$

As a result, the maximum distance between two consecutive quantum points is equal to $(1 + \sqrt{5})r_{QD}$. If the distance between two consecutive quantum dots is greater than this value, then the applied force between the two quantum dots causes torque and, eventually, the sheet bends between the two. Additionally, $\Delta$ must always be positive, so that the

minimum distance ($\Delta_{\min}$) of consecutive dots is equal to $2r_{QD}$. All in all, the distances that consecutive quantum dots can have are equal to $2r_{QD} \leq |z_{Qi} - z_{Qi-1}| \leq (1 + \sqrt{5})r_{QD}$.

On the other hand, when knowing the exact distance between consecutive quantum dots and assuming that the distance between all quantum dots is the same, their number can be easily given, as follows:

$$n = \Im\left(\frac{L_z}{r_{QD}}\right) + 1, \quad 2 < \Im^{-1} < 1 + \sqrt{5} \tag{14}$$

So far, we have been able to present how to calculate the amount of force induced on a charged sheet of graphene nanoribbon by the array of quantum dots, as well as how to arrange and number the quantum dots. We find the ratio of the length of the graphene nanoribbon to the radius of the quantum dot important. Formula (7) can be converted to an integral in the z-direction if this ratio is too large ($L_z/r_{QD} \gg 1$); as a result, the inductive force on the charged sheet of graphene nanoribbon is given as follows:

$$\vec{F} = \frac{3(y_p - r_{QD})q \cdot \tilde{p}_0^{QD}}{4\pi^2\varepsilon_0}\left(\frac{\mu_c}{\hbar v_f}\right)^2 \int_0^{L_z} \frac{dz}{R^3}\left[\frac{x_p}{R^2}\quad\left(\left(\frac{y_p - r_{QD}}{R^2}\right) - \frac{1}{3(y_p - r_{QD})}\right)\quad \frac{z}{R^2}\right]\begin{bmatrix}\cos\varphi & -\sin\varphi & 0\\\sin\varphi & \cos\varphi & 0\\0 & 0 & 1\end{bmatrix}\begin{bmatrix}a_r\\a_\varphi\\a_z\end{bmatrix} \tag{15}$$

where $R = \sqrt{x_p^2 + (y_p - r_{QD})^2 + z^2} \Rightarrow dR = \frac{z}{R}dz, y_p = 0, x_p = x_g$.

The result of the integration under these conditions is the final inductive force at any point on the charged graphene nanoribbon sheet, which follows the following equation:

$$\vec{F} = \mathbf{D.H.}\begin{bmatrix}\cos\varphi & -\sin\varphi & 0\\\sin\varphi & \cos\varphi & 0\\0 & 0 & 1\end{bmatrix}\begin{bmatrix}a_r\\a_\varphi\\a_z\end{bmatrix}$$

$$\mathbf{D} = \frac{3q \cdot \tilde{p}_0^{QD}}{4\pi^2\varepsilon_0}\left(\frac{\mu_c}{\hbar v_f}\right)^2 = \frac{6|q|^2(\varepsilon_r - 1)}{\varepsilon_0\sqrt{L_x^2 + L_z^2}}\left(\frac{\mu_c}{2\pi\hbar v_f}\right)^4 \tag{16}$$

$$\mathbf{H} = \left[\frac{x_g L_z r_{QD}\left(3R^2|_{z=L_z} - L_z^2\right)}{R^3|_{z=L_z}\left(x_g^2 + r_{QD}^2\right)^2} \quad \frac{L_z r_{QD}^2\left(3R^2|_{z=L_z} - L_z^2\right)}{R^3|_{z=L_z}\left(x_g^2 + r_{QD}^2\right)^2} - \frac{L_z r_{QD}}{3R|_{z=L_z}\left(x_g^2 + r_{QD}^2\right)} \quad \frac{r_{QD}}{4}\left(\frac{1}{\left(x_g^2 + r_{QD}^2\right)^{\frac{3}{2}}} - \frac{1}{\left(x_g^2 + r_{QD}^2 + L_z\right)^{\frac{3}{2}}}\right)\right]$$

where $\mathbf{D}$ is a coefficient proportional to the length and width of the graphene nanoribbon as well as the permeability of the quantum dots. Therefore, by changing $\mathbf{D}$, their physical characteristics will change. The matrix $\mathbf{H}$ shows the force applied to any point of the charged graphene nanoribbon sheet in three directions x, y, and z.

*3.3. Application of Electromagnetic Waves to Control the Bending Rate of the Charged Sheet of Graphene Nanoribbon*

Using an array of quantum dots with the same radius and distance from one another, on the longitudinal axis that passes through the center of the charged sheet of graphene nanoribbon, we were able to bend it. However, this bending must be managed in such a way as to create a graphene nanotube (SWCNT). We performed bending management with the help of electromagnetic wave applications. Therefore, the purpose of this section is to design the appropriate amplitude for electromagnetic waves. The charged sheet of graphene nanoribbon bends under force applied from the side of the quantum dot array to a certain point ($X_{gi}, Y_{gi}, Z_{gi}$). According to Figure 5, to achieve the SWCNT structure, it is necessary to design a force that can cause the sheet to bend in the path of the vector $\Delta L$, which indicates the length of the sheet displacement.

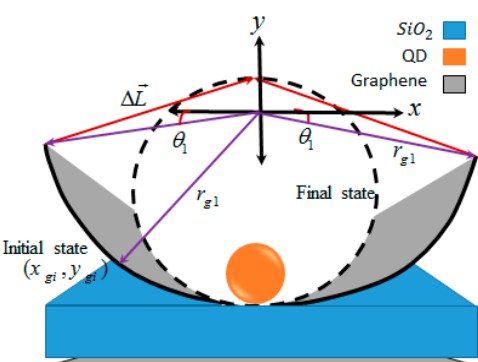

**Figure 5.** Introduction of the vector $\Delta L$. In addition, schematic representation of the cross-section of the SWCNT, and visual expression of the bending management of the charged graphene nanoribbon sheet with the help of the bending vector $\Delta L$.

The function that follows the circular cross-section of the SWCNT is given by the equation $x^2 + y^2 = (L_x/2\pi)^2$. Accordingly, the displacement vector ($\Delta L$) in the bend is given as follows:

$$\Delta \vec{L} = \left( \sqrt{\left( \frac{L_x}{2\pi} \right)^2 - y^2} - x_{g1} \right) a_x + (y - y_{g1}) a_y \qquad (17)$$

According to Figure 5, we can obtain the electric field emitted from the various states ($E^g_{yi}$) of the graphene nanoribbon sheet—which varies from the initial state to the final—using Gauss'law [41]. In the initial stages, the angle $\theta_1$ could be varied between $-\frac{\pi}{4}$ and $\frac{\pi}{2}$. A bending angle ($\theta_s$) similar to the angle $\theta_1$ (see Figure 5) was defined for the changing states of the charged graphene nanoribbon sheet. Ultimately, the electric field emitted from different states of graphene nanoribbons exists in the quantum dots, and is given as follows [41]:

$$E^g_{yi}(\theta_s) = \frac{Q}{4\pi\varepsilon_0} \left( \frac{\pi + 2\theta_s}{L_x \sqrt{L_s^2 + r^2}} \right) \sin\phi \Bigg|_{\phi = \frac{\pi}{2}, r = r_{QD}} \approx \left( \frac{\pi + 2\theta_s}{4\pi\varepsilon_0} \right) \left( \frac{Q}{L_x L_z} \right) \qquad (18)$$

According to Lorentz's force law, if we apply waves with a certain amplitude and phase to the system, a force is applied to the charged graphene nanoribbon sheet [42] (Zhang, 1998 #117). This force is given as follows:

$$\vec{F} = Q \cdot \vec{E} + Q \vec{v} \times \vec{B} \qquad (19)$$

Since the system does not have a closed path to establish current, the magnetic field does not affect the operating waves of the system, and the magnitude $qv \times B$ is zero (the speed of movement of the charged object = $\vec{v}$). If the electric field of the applied waves is in the direction of the SWNCT's radius ($E_r = E_i \cos(\omega t - kz)$), the polarization of the quantum array changes to the following form:

$$\tilde{p}^{QD} = \varepsilon_0 (\varepsilon_r - 1) \left( E^g_{yi} + E_i \cos\alpha \right) a_y \qquad (20)$$

Formula (20) is obtained by assuming that the interaction of quantum arrays with one another is not considered. The polarization of the quantum dot array must be zero before the desired cross-sectional area of the SWCNT is created. Moreover, the array of quantum dots should not be polarized in the x-direction with the applied waves, so 'sinα' is zero at the pivot points. As a result, the electric field of the applied waves is in the y-direction. Finally, the polarization of the quantum dots must change to neutralize the effect of the

electric field force of the applied waves on the lateral surface of the SWCNT. As a result, the amplitude of the applied waves is given as follows:

$$E_i = -\frac{E_{yi}^g}{\varepsilon_0(\varepsilon_r - 1)} = -\left(\frac{\pi + 2\theta_S}{4\pi\varepsilon_0^2(\varepsilon_r - 1)}\right)\left(\frac{Q}{L_x L_z}\right) \tag{21}$$

*3.4. Application of DC Voltage to Control the Bending Rate by Adjusting the Charge Rate of the Graphene Nanoribbon Sheet*

From Section 3.2, we can conclude that the force exerted by the polarization of quantum dots causes the graphene nanoribbon sheet to bend. In Section 3.3, we attempted to direct this bending with electromagnetic waves directed toward the nanotube. In addition, we found the importance of the magnitude of the applied voltage to the initial bending rate. If the charge of the graphene nanoribbon sheet increases, the field emitted from it increases as well. Eventually, increasing the polarization rate of the quantum dot array will result in the force applied to the surface of the graphene nanoribbon sheet being greater, and the initial bending angle being larger. Through Equation (6), we can find the relationship between the charge of a graphene nanoribbon sheet and the polarization of a quantum array. Concerning Gauss's law and $\frac{L_z}{r_{QD}}, \frac{L_x}{r_{QD}} >> 1$:

$$\widetilde{p}^{QD} = \frac{qn_s(\varepsilon_r - 1)\Delta\varphi}{4\pi L_z L_x r_{QD}}, \quad \Delta\varphi = \pi + 2\theta_1 \tag{22}$$

where $\theta_1$ equals $\cos^{-1}\left(\frac{\pi}{\sqrt{1+\pi^2}}\right)$ when the sheet of graphene nanoribbon is one plate. This changes with an increase in DC voltage. Therefore, under these assumptions, the field emitted from the quantum dots, as expressed in Equation (7), can be obtained as follows:

$$E^{arry \quad Dipolar} = \frac{\widetilde{p}^{QD}}{4\pi\varepsilon_0 r^3}\begin{bmatrix}\sin\varphi & \cos\varphi & -\sin\varphi\end{bmatrix}\begin{bmatrix}a_r \\ a_\varphi \\ a_z\end{bmatrix} \tag{23}$$

On the other hand, according to the structure of Figure 1, graphene nanoribbon plates and the material between them create a single capacitive charge that is stored on the plate, which is equal to $Q = C.V_{DC}$ [40, 41]. Capacitance is determined based on the introduced structure ($C = \varepsilon_0\varepsilon_{sio2}L_xL_z/d$). Moreover, it is sometimes expressed in surface units, which indicate the amount of charge stored per unit of area ($C = \varepsilon_0\varepsilon_{sio2}L_xL_z/d$) ($C = \varepsilon_0\varepsilon_{sio2}/d$). Therefore, if we want to calculate the potential energy in the graphene sheet, we can use the capacitance per unit of area, because we divide the graphene sheet into small integrated plates with an area of 1 nm$^2$ to calculate its displacement relative to the applied energy from the quantum array. In this case, the surface charge of graphene is proportional to its surface conductivity ($Q_s = \sigma_s$). The reason for this importance is that, when the charged test object is moved in the field by some external agent, the work done by the field on the charge is equal to the negative of the work done by the external agent causing the displacement. This work depends only on the particle's initial and final coordinates. Hence, we can derive Equation (24):

$$\Delta U = \vec{F}\cdot\vec{\Delta R} = -\sigma\int_A^B \vec{E}^{arry \quad Dipolar}\cdot d\vec{s} \tag{24}$$

where '$F$' represents the result of forces applied to each surface unit of a graphene sheet and displaced by $\Delta R$, '$A$' is the state where the graphene nanoribbon sheet is a plate shape, and '$B$' is when the graphene nanoribbon sheet forms the SWCNT. As a result, the potential energy stored in the structure during the work is given as follows:

$$\Delta U \approx 3V_{DC}\left(\frac{\sigma_s\varepsilon_{sio2}}{d\cdot r_{QD}(4\pi)\varepsilon_0}\right)L_z \tag{25}$$

The force applied to the graphene nanoribbon sheet's edges is proportional to the applied voltage. Moreover, we can adjust the force required to create the SWCNT structure by adjusting the voltage. Thus, without applying electromagnetic waves, we can achieve the following change in the structure of our system:

$$\vec{F} \approx 3V_{DC}\left(\frac{\sigma_s \varepsilon_{sio2}}{d \cdot r_{QD}(4\pi)\varepsilon_0}\right)L_z\left(\frac{\vec{\Delta R}}{\left|\vec{\Delta R}\right|^2}\right), \quad \vec{\Delta R}_{\text{edges}} = \left(\frac{L_x}{2}\right)\left(a_x + \left(\frac{1}{\pi}\right)a_y\right) \quad (26)$$

## 4. Results and Discussion

In this work, a charged sheet of graphene nanoribbon was converted to an SWNCT using an array of quantum dots and electromagnetic waves. Numerous applications for this can be imagined in the field of optics. We used an array of quantum dots as a pivot in the beginning. Experiments showed that quantum arrays on the graphene nanoribbon cause the graphene to bend. We analyzed this using the laws of classical physics. Then, we tried to manage the bending of the graphene nanoribbon sheet by applying electromagnetic waves in a suitable direction and amplitude. As a result, the structure of graphene nanotubes was theoretically obtained. We simulated the proposed structure numerically with arbitrary values. Based on this, we can see the force that is applied to each section ($z = z_{gi}$) of the graphene nanoribbon sheet through the array of quantum dots in three directions, as shown in Figure 6. The magnitude of this force is expressed in Equation (16).

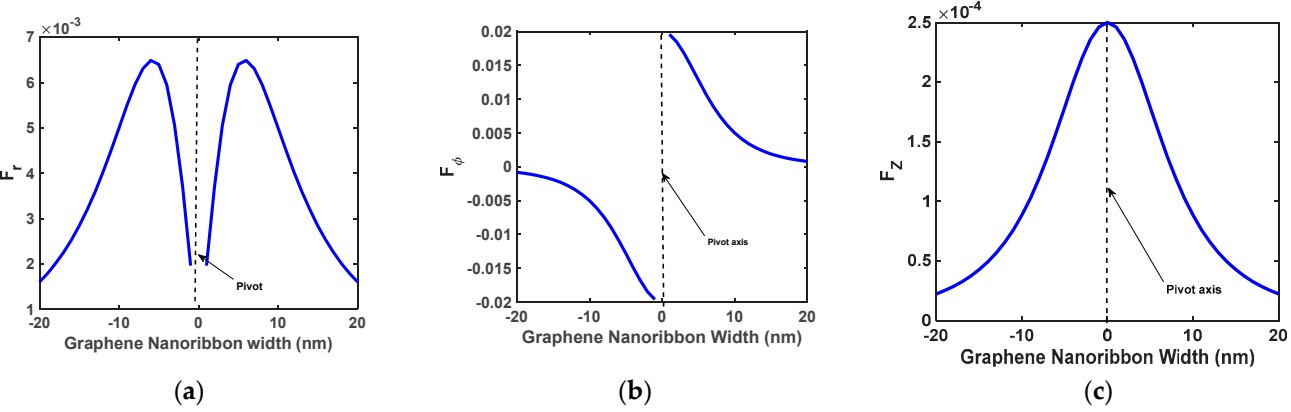

**Figure 6.** The force applied to each section ($z = z_{gi}$) of the graphene nanoribbon sheet through the quantum array in the (**a**) *r*-direction, (**b**) $\phi$-direction, and (**c**) *z*-direction.

In Figure 6, the forces are symmetrical, so they cause symmetrical displacement on both sides of the pivot axis. There is a break in Figure 6 around the pivotal axis due to the fact that in order to create a cylindrical structure without defects, the first condition is that the force applied to the pipe axis must be zero, so the total electric field effective on this axis must also be zero. As we know, if the length of the cylinder is in the z-direction, a cylinder has axes of symmetry in the plane y = 0 or x = 0. In this work, the axis of the considered pivot was located on the plane x = 0. This indicates symmetrical bending on both sides of the pivot. As a result, according to the diagrams presented in Figure 6, the presented computation and design are validated theoretically [11,16]. The charged sheet of graphene nanoribbon emits electric waves that cause the array of quantum dots to be polarized. The polarization of the quantum dots exerts the same force on each surface unit of a charged sheet of graphene nanoribbon with equal coordinates on the XY plane, so that each surface unit of the sheet bends symmetrically into both pivot sides to location r at an angle of θ. In future works, the amount of bending could be theoretically calculated using the quantum physics of the particles and the force applied to each particle in the

structure of the graphene nanoribbon sheet, or the amount of bending could be reported with the help of experiments in the fabrication of this graphene hybrid state. In the next step, we used electromagnetic waves to control the continued bending of the charged sheet of graphene nanoribbon. In Figure 7, we can see the dependence of the amplitude of the electromagnetic waves applied by the electric field on the angle and location of the charged graphene nanoribbon sheet.

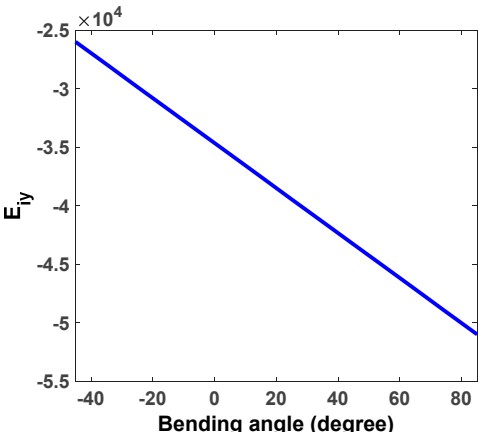

**Figure 7.** Dependence of the amplitude of the electromagnetic waves applied by the electric field on the bending angle.

The bending angle is the angle at which the edge of the graphene sheet meets with its bending center. This means that each time the $\theta_s$ changes value, the graphene sheet becomes more curved and induces more electric fields at the quantum dots. As a result, to counteract this effect on the quantum dots, it is necessary to increase the amplitude of the applied waves in the opposite direction of the electric field induced by the bent graphene sheet. The physical references that report the electric field emitted from the charged curved plates in their surroundings have values similar to those shown in Figure 7 at the specified angle. For example, the location of the pivot (bent on the charged sheet) when the bending angle is 90 degrees (a cylinder with a radius $\frac{L_x}{2\pi}$) is equal.

According to the work–energy law, we can express the relation of the force exerted by the polarization of the quantum dots in terms of externally applied voltage. Figure 8 shows this relation.

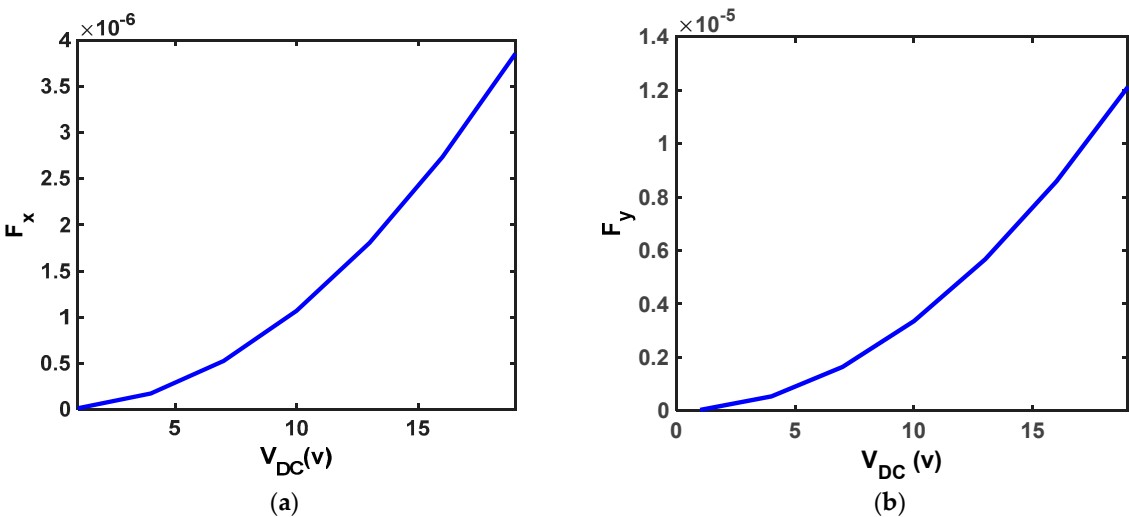

**Figure 8.** The relationship between the applied voltage and the force applied in the (**a**) x-direction and (**b**) y-direction to a graphene nanoribbon sheet (d = 25 nm).



As expected, the relationship between the force and the applied voltage is a quadratic power law. This approach is correct based on the basics of physics, because the energy stored in the capacitor is ($W = (1/2)CV^2$).

In synthesizing graphene with excellent materials, different experimental methods have been compared in different studies, as well as in the manufacture of carbon nanotubes. These reports are only based on laboratory data, and we have not found any theoretical analysis of the amount of bending. The previous samples were compared based on the type of material, experimental technique, tensile strength, strain, etc., but unfortunately only from a laboratory perspective [26,30]. The proposed structure in the [43] defines the up-conversion process of the optical force proposed and designed to drive a nanoscale optical mechanical structure. This structure is presented as an optical detector and a high multi-wavelength conversion system. The final advantage of this system is photodetection at room temperature even for long wavelengths. On the other hand, the main weakness of hybrid graphene-based infrared photodetector with engineering of trap levels using organic molecules operating at room temperature, and with a fast response time compared to other detectors, is related to 2D electron gas located at the surface—these electrons react to the surrounding atmosphere, and the system is affected by the surrounding gases. Therefore, the high conductivity of graphene results in a high dark current for these detectors. The structure of these detectors overcomes this problem [30]. Both of these structures have been introduced to electro-optical structures using nanostructure bending that has been practically observed in the laboratory. Therefore, the need for theoretical data to reduce the error and the number of experiments required to achieve the desired result confirms the value of this work.

## 5. Conclusions

In this work, we merged the physics of electricity, mechanics, and electromagnetics to explain how to bend a sheet of a graphene nanoribbon using an array of quantum dots. We charged the graphene sheet with externally applied DC voltage, causing the array of QDs to polarize. The induced electric field from the polarized QDs forced the charged graphene sheet to bend. To precisely form the SWCNT structure, we needed to manage the bending correctly, which was achieved using externally applied electromagnetic waves. It should be noted that the correct choice of the array of QDs and the presence of dangling bonds in the graphene nanoribbon were used to create the nanotube. The main purpose of this work was proper management of the devices at the nano scale for various optical and electrical applications. To illustrate the idea, we used the graphene nanoribbon's unique properties to make SWCNTs. This is an incredibly pristine presentation that can change the geometric structure of devices by applying electromagnetic waves. As we know, changing the geometry of a device changes its physical properties ($\mu, \varepsilon, \sigma$). Therefore, a device with two or more applications can be designed. Furthermore, this opens new possibilities for the design and manufacture of electro-optical and optical devices.

**Author Contributions:** S.A. did numerical simulation and modeling, A.R. and P.M. supervised the project, conceptualized and revised and rewrite the paper. All authors have read and agreed to the published version of the manuscript.

**Funding:** This research received no external funding.

**Institutional Review Board Statement:** Not applicable.

**Informed Consent Statement:** Not applicable.

**Data Availability Statement:** Data sharing is not applicable to this article.

**Conflicts of Interest:** The authors declare no conflict of interest.

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
