# Peer review of "Graphene Nanoribbon Bending (Nanotubes): Interaction Force between QDs and Graphene"

_coatings, doi:10.3390/coatings12091341_

Round 1
Reviewer 1 Report
The article can be accepted in the present form subjected to the following minor revisions:
1. The following relevant articles can also be cited for the sake of completeness: https://doi.org/10.1016/j.tws.2021.108626, https://doi.org/10.1007/s11831-021-09652-0, https://doi.org/10.1080/15397734.2021.1977659
2. Minor grammatical and spelling errors exist.
3. Why there is a break in fIg. 6 about the pivotal axis?
Author Response
Dear Editor
Enclosed is the revised version of our manuscript and submitted for your consideration. We addressed all comments in the paper and modified the paper and in the following, a short answer for reviewers is presented.
Bests
Ali Rostami

Reviewer 2 Report
The start with the general comment is to read carefully the manuscript and try to eliminate repetitions, check the occurrence of capital letters, etc. Some points/examples or errors or inconsistencies are as follows:
Among the methods of obtaining graphene also the production of such material on a substrate of a metallic liquid phase or CVD techniques should be mentioned.
"However, the request for increasing the stability of the building element without an increase in the size of the elements leads to the introduction of reinforcements." - the sentence is unclear.
The additional reference proving the predominant usage of SWCNT in biomedicine is necessary.
In line 87 authors state that graphene "has just been invented" which may be too big a simplification. Not only it is about 20 years since graphene's beginnings but also we are now focusing on new graphene-based materials like spatial graphene. Please rewrite that part.
In line 107 please change "reference" to Siahsar, M., et al. or rephrase.
Why the word "de-turn" was used/created in line 110? Why not use standard nomenclature there?
Lines 368-369 need re-writing because of Experimental experiments, capital letters inside the sentences, and a general lack of sense in the context of the manuscript
The work-Energy Law is rather Work-Energy Theorem.
As far as Chapters 3 and 4 are considered there is no clear division between mathematical calculations/simulations and experimental results. It would be perfect to clearly state it from the beginning in the Materials and Methods part. It would be also a place for information such as 373-374 (in fact those are not the results but the experimental setup).
More basic information concerning the materials and apparatus will be perfect for example:
What was the origin of quantum dots and graphene nanoribbon sheet (self-synthesized - in what parameters, bought from whom? What was an experimental setup for applying electromagnetic waves? etc.
Finally, the manuscript deals with an interesting topic so to my mind it is worth publication yet it requires further work to improve it to such extent that will ensure both its clarity and high scientific impact
Author Response

(The authors gave the same response as above.)

Reviewer 3 Report
I carefully read the paper. It is a kind of perfect work. The idea is new in the field and it is of practical applications. However, some minor changes must be taken into account as follows
1. Line 315, please change the word (the sentence) by (the magnitude).
2. Please revise the paper regarding grammar.
3. Although the results seem good, there is no comparisons with previous.
4. The paper may be a new idea to merge the physics of electricity, mechanics, and electromagnetics to explain how to bend a sheet of graphene nanoribbon using an array of quantum dots. The authors must validate their results. Such issue is essential for other future papers which will extend the concept proposed in this work.
Author Response

(The authors gave the same response as above.)

Round 2
Reviewer 2 Report
The manuscript is more less ready to published
Best wishes